# Analysis of the Azimuth Ambiguity and Imaging Area Restriction for Circular SAR Based on the Back-Projection Algorithm

**DOI:** 10.3390/s19224920

**Published:** 2019-11-12

**Authors:** Bang Du, Xiaolan Qiu, Lijia Huang, Songlin Lei, Bin Lei, Chibiao Ding

**Affiliations:** 1Aerospace Information Research Institute, Chinese Academy of Sciences, Beijing 100094, China; dubang17@mails.ucas.ac.cn (B.D.); iecas8huanglijia@163.com (L.H.); leisonglin1913@163.com (S.L.); leibin@mail.ie.ac.cn (B.L.); cbding@mail.ie.ac.cn (C.D.); 2School of Electronic, Electrical and Communication Engineering, University of Chinese Academy of Sciences, Beijing 100049, China; 3Key Laboratory of Technology in Geo-spatial Information Processing and Application Systems, Chinese Academic of Science, Beijing 100190, China; 4Institute of Electronics, Chinese Academy of Sciences, Beijing 100190, China

**Keywords:** circular SAR, azimuth ambiguity, BP algorithm, synthetic aperture radar imaging

## Abstract

Circular synthetic aperture radar (CSAR) has a 360° observation capability on the central observation scenario. A typical way to process CSAR imaging is to cut data into small sub-apertures because most targets are only coherent at a very small observation angle. There are many sub-aperture imaging methods after development in recent years. The back-projection algorithm is widely used because it is simple and can be applied to an arbitrary trajectory. Because of the limitation of the Nyquist sampling frequency and influence of the antenna sidelobe, azimuth ambiguity is a phenomenon that may occur in the radar imaging process. The existing researches typically choose the back-projection (BP) imaging area according to the SAR platform flight path and the antenna beam width. The limitation of the CSAR imaging area and its azimuth ambiguity region are rarely analyzed theoretically. This paper focus on the sub-aperture imaging of CSAR, based on the BP algorithm, which derives the relationship of azimuth ambiguity with CSAR parameters such as the pause repeat frequency (PRF), slant range angle, velocity of radar platform, etc. This paper proposes an equation for the calculation of the azimuth ambiguity region and analyzes the limitations, which provides theoretical support for CSAR parameter design, imaging area selection, and azimuth ambiguity analysis.

## 1. Introduction

Circular synthetic aperture radar (CSAR) is a special imaging mode of SAR whose radar platform moves along a circular track. Compared with traditional SAR, CSAR can provide higher resolution and all-around information on targets. Additionally, CSAR has the potential for three-dimensional imaging [1]. Due to these unique properties different from linear path SAR, CSAR receives wide attention. Several institutes around the world, such as the American Air Force Research Lab [2,3], German Aerospace Center (DLR) [4,5], French Aerospace Center (ONERA) [6], and Institute of Electrics, Chinese Academy of Sciences [7], have done experiments on CSAR.

There are several imaging methods for CSAR imaging. The frequency-domain processor, based on the analysis of the three-dimensional Green’s function, is proposed [8], which is computational and efficient. However, this algorithm is under the assumption that the platform trajectory is well shaped and the imaging surface is flat, which is hard to be applied. Time-domain processing using the back-projection (BP) algorithm [9], unlike the frequency-domain algorithm, can apply to arbitrary trajectory, which is widely used. Furthermore, the fast-factorized back-projection algorithm is presented to reduce the computational burden of the time-domain process in [10,11].

Because of the limitations of the Nyquist sampling theorem, the azimuth ambiguity phenomenon may also occur during CSAR imaging. In the past few years, the theoretical study on CSAR emphasizes a concentric circle containing the center of the scene, which is in the 360° beam irradiation. In this assumption, pause repeat frequency (PRF) is supposed to be larger than the instant Doppler bandwidth of the concentric circle [12]. Actually, the imaging area of each sub-aperture is usually larger than this circle area, and because of the beam control error and the attitude measurement error of the SAR, the imaging area setting by the BP algorithm needs some redundancy. However, if the BP imaging area is too large, there will be azimuth ambiguity. The target illuminated by the antenna sidelobes will also cause azimuth ambiguity in the central imaging area, and vice versa. Using the back-projection algorithm may produce azimuth ambiguity for larger areas, including these targets. Theoretical research on these issues is rarely mentioned in the existing literature.

This study, we focus on the azimuth ambiguity of CSAR and analyze the relationship between azimuth ambiguity and CSAR parameters, providing the expression for the ambiguity area based on the back-projection imaging algorithm. In addition, we provide a better parameter design method that can offer an arbitrarily selected area in the entire reference height plane and avoid azimuth ambiguity.

The remainder of the article is organized as follows: In Section 2, azimuth ambiguity is analyzed based on the brief introduction of the CSAR imaging model, and formulations of imaging area center causing ambiguity is given. Ideal point target simulation combined with a real CSAR data experiment is given in Section 3. The results are discussed and summarized in Section 4.

## 2. Materials and Methods

### 2.1. CSAR Signal Model

An ideal imaging geometry is shown in Figure 1. Assume the SAR platform moves with the angular velocity ω along an ideal anti-clockwise circle whose radius is L and height is H. An arbitrary target located in the beam illuminating area can be expressed as $\left(x_{0},y_{0},z_{0}\right)$. In addition, assume the radar is side looking and $R_{S}$ starts flying at (0,L,H) for the simplicity of analysis. Thus, the instantaneous distance of an arbitrary target shown in Figure 1 at any slow time t can be expressed in a simple form:
(1)$$R\left(t\right)={\left(Lcos\omega t-x_{0}\right)}^{2}+{\left(Lsin\omega t-y_{0}\right)}^{2}+{\left(H-z_{0}\right)}^{2}.$$

According to [13], after demodulation and range compression, the received linear frequency modulated (LFM) signal of a target can be represented by Equation (2):(2)$$S\left(t,\tau \right)=\sigma S_{rc}\left(\tau -\frac{2R\left(t|x_{0},y_{0}\right)}{c}\right)exp\left(-j4\pi \frac{R\left(t|x_{0},y_{0}\right)}{\lambda }\right)$$
where σ is the backscattering coefficient of the point, $S_{rc}$ is the pulse after range compressing, τ is the fast time, c is the velocity of light, and λ is the wavelength. Assuming the imaging grid of the BP algorithm is a Cartesian grid denoted as (m,n), according to the Equation (2), the value of the targets after focusing is [9]:(3)$$I\left(m,n|x_{0},y_{0}\right)=\int S\left(t,\tau \right)exp\left(j4\pi \frac{R(t|x_{0},y_{0})}{\lambda }\right)dt.$$

This is the basic formulation of the back-projection algorithm. The height of the Cartesian grid can be formed by the digital elevation model (DEM) or just given by the reference height plane.

### 2.2. Azimuth Ambiguity Analysis

Consider an arbitrary point target $\left(x_{0},y_{0},z_{0}\right)$ located in the scene. At any slow time in a sub-aperture, the generated false point $\left(x^{\prime },y^{\prime }\right)$ using the back-projection algorithm at a reference height $H_{ref}$ satisfies the following constraint:(4)$$R\left(t,x^{\prime },y^{\prime },H_{ref}\right)=R\left(t,x_{0},y_{0},z_{0}\right).$$

In the CSAR geometry, there is not any exact solution for $\left(x^{\prime },y^{\prime }\right)$ to satisfy the above equation. However, if the target $\left(x^{\prime },y^{\prime }\right)$ also satisfy the following equation, which is
(5)$$F_{dc}\left(x^{\prime },y^{\prime },H_{ref}\right)=kPRF+F_{dc}\left(x_{0},y_{0},z_{0}\right)\;\;k=0,\;\pm 1,\;\pm 2\ldots ,$$
then, it still will cause ambiguity. In the above equation, $F_{dc}$ is the Doppler frequency and PRF is the pulse repetition frequency. Equations (4) and (5) are based on the assumption that there are points that have a similar distance history. According to the basic radar principle, the $F_{dc}$ of point $\left(x_{0},y_{0},z_{0}\right)$ at slow time t is
(6)$$F_{dc}=-\frac{2}{\lambda }\frac{dR\left(t\right)}{dt}=-\frac{2}{\lambda }\frac{L\omega \left(x_{0}sin\omega t-y_{0}cos\omega t\right)}{\sqrt{{\left(Lcos\omega t-x_{0}\right)}^{2}+{\left(Lsin\omega t-y_{0}\right)}^{2}+{\left(H-z_{0}\right)}^{2}}}.$$

Then, we consider the center point (0,0,0) as an example to simplify the analysis. Set the reference height plane as zero for the purpose of finding points of the same height which fit the constraint, and Equations (4) and (5) turn into the following expression after simplification:(7)$${\left(Lcos\omega t-x^{\prime }\right)}^{2}+{\left(Lsin\omega t-y^{\prime }\right)}^{2}=L^{2},$$
(8)$$x^{\prime }sin\omega t-y^{\prime }cos\omega t=-kPRF\frac{\lambda }{2L\omega }\sqrt{L^{2}+H^{2}\;}\;\;k=0,\pm 1,\pm 2\ldots$$

Notice that $kPRF\frac{\lambda }{2L\omega }\sqrt{L^{2}+H^{2}}\left(\;k=0,\pm 1,\pm 2\ldots \right)$ is a constant that is determined by the CSAR parameter. Let $-kPRF\frac{\lambda }{2L\omega }\sqrt{L^{2}+H^{2}}=A$, simultaneous Equations (7) and (8), and finally we can get a quadratic function with one unknown:(9)$${x^{\prime }}^{2}-2x^{\prime }\left(Lcos\omega t+Asin\omega t\right)+A^{2}+2ALsin\omega tcos\omega t=0.$$

Solve Equation (9) and substitute the solution $x^{\prime }$ into Equation (8). Then, the formulation of the wrong points corresponding to the center point (0,0,0) generated by the back-projecting algorithm on reference height plane z=0 is
(10)$$\left(Lcos\omega t+Asin\omega t\pm \left|cos\omega t\right|\sqrt{L^{2}-A^{2}},\;Lsin\omega t-Acos\omega t\pm \left|sin\omega t\right|\sqrt{L^{2}-A^{2}}\right).$$

Furthermore, points on the straight line y=xtanωt can also be deduced in a similar way after observing Equations (7) and (8). If we replace y by rsinωt, and replace x by rcosωt, in which r represents the distance to the origin. Then L in Equations (7) and (8) can be replaced by ${\left(L-r\right)}^{2}$. Obviously, we are able to get a more general expression using the same process for points (rcosωt,rsinωt) for each exact slow time:(11)$$\left(Lcos\omega t+A^{\prime }sin\omega t\pm \left|cos\omega t\right|\sqrt{{\left(L-r\right)}^{2}-{A^{\prime }}^{2}},Lsin\omega t-A^{\prime }cos\omega t\pm \left|sin\omega t\right|\sqrt{{\left(L-r\right)}^{2}-{A^{\prime }}^{2}}\right),$$
where $A^{\prime }$ equals to $-kPRF\frac{\lambda }{2L\omega }\sqrt{{\left(L-r\right)}^{2}+H^{2}}$, which is a constant. Treat r as an unknown variable and start the elimination; points on the line y=xtanωt form a hyperbolic curve of fuzzy point collection at any slow time t. The formula of the hyperbolic curve is
(12)$$\left(1-M^{2}\right)P^{2}-Q^{2}=H^{2}.$$

In Equation (12), $P=\frac{x^{\prime }sin\omega t+y^{\prime }cos\omega t}{M}$, $Q=x^{\prime }cos\omega t+y^{\prime }sin\omega t-L$, and $M=\;-kPRF\frac{\lambda }{2L\omega }$. The complete derivation for Equation (12) can be found in Appendix A.

### 2.3. Imaging Range Limitation

Here we focus on the constant k and its influence on imaging area and fuzzy point quality.

If k=0, Equation (10) can be simplified as (2Lcosωt,2Lsinωt), the exact symmetric point of (0,0,0) of the trajectory. When slow time t changes in a sub-aperture $-\frac{T}{2}:\frac{T}{2}$, this symmetric point moves as a circular arc whose radius is 2L. Thus, the symmetric point is defocused into an area with a certain amount of energy after back-projection processing near the point. This phenomenon can be easily avoided because the antenna’s looking direction can be determined.

If k≠0, due to the symmetry of the CSAR trajectory, first take $\omega t\in \left(0,\frac{\pi }{2}\right)$ into account. Notice that sinωt,cosωt>0 in the assumption, and we ignore the point outside of the circular trajectory for the same reason. Obviously, $L-\sqrt{L^{2}-A^{2}}$ is a constant number that can be replaced by B. Then Equation (10) can be simplified as (Bcosωt+Asinωt,−Acosωt+Bsinωt); this fuzzy point $\left(x^{\prime },y^{\prime }\right)$ moves along an ellipse curve when t changes. The formula of the ellipse curve is shown below, and this derivation can be found in Appendix B:(13)$${\left(\frac{Ax^{\prime }-By^{\prime }}{A^{2}-B^{2}}\right)}^{2}+{\left(\frac{Ax^{\prime }+By^{\prime }}{A^{2}+B^{2}}\right)}^{2}=1.$$

Next, we consider $\omega t\in \left(\frac{\pi }{2},\pi \right)$. We can still let $B=L-\sqrt{L^{2}-A^{2}}$ and then replace $L+\sqrt{L^{2}-A^{2}}$ by C; the simulation process is similar to Appendix B, and we can get another formula of the ellipse curve:(14)$${\left(\frac{Ax^{\prime }-By^{\prime }}{A^{2}-BC}\right)}^{2}+{\left(\frac{Ax^{\prime }+By^{\prime }}{\left(A^{2}+BC\right)}\right)}^{2}=1,$$
because we are really concerned about the points that might locate in the beam irradiation and get mixed with real targets. Though there are two solutions for every k≠0, only one solution is adopted. Furthermore, these fuzzy points are defocused and formed into an area on the account that the distance history is equal only at the exact slow time t. To avoid the occurrence of this phenomenon violently, we can let Δ <0, then the quadratic Equation (9) has no solution, which means
(15)$$PRF>\frac{2\omega L^{2}}{\lambda \left|k\right|\sqrt{L^{2}+H^{2}}},k=\pm 1,\pm 2\ldots$$

### 2.4. Comparison with the Traditional Method

In the traditional method [8], an area irradiated by the full aperture is taken into consideration. The radius $R_{s}$ is mainly determined by the azimuth beam width. According to the fundamental theory, PRF should be larger than the bandwidth, which is formed by the instant Doppler frequency of all targets in the area. This theory is applicable for the strip-map and spotlight radar. In CSAR mode, the formula is based on the Nyquist sampling theorem [8]. We transform the expression using the variable named in Figure 1:(16)$$PRF>\frac{2\omega LR_{s}K_{max}}{\pi \sqrt{L^{2}+H^{2}}}$$
where $R_{s}$ is the radius of the image region, and $K_{max}$ is the max wave number in the illuminated area. Choose |k|=1 in Equation (15), then the formula is similar to Equation (16). However, azimuth ambiguity of the center point (0,0,0) can be completely eliminated. In other words, the phenomenon of azimuth ambiguity using the back-projection algorithm for point (0,0,0) cannot be found at reference height. However, the analysis method in the article can explain the phenomenon of azimuth ambiguity and the effect on the main lobe of the strong scattering point located in the sidelobe.

## 3. Experimental Results

### 3.1. Ideal Point Simulation

This section we first take the ideal point at (0,0,0) into simulation. The main parameters are shown in Table 1.

First, substitute the parameters for simulation to Equation (9); k=±1,±2 all satisfy the judgment of the existing solution. Then, calculate each fuzzy point using Equation (10). We are only concerned about points, which are located inside of the platform trajectory, because these points may be irradiated by radar. Figure 2 shows the geometry of the back-projection on the plane z=0. It can be discussed that one of the ambiguity points may be possible in the whole sub-aperture beam irradiation. Figure 3 shows the ideal point simulation in three areas using the BP algorithm. While for images on the ambiguity areas, the whole sub-aperture is used, ignoring the limitation of beam irradiation, and the size of the imaging area is adjusted to cover the whole defocused point. It can be seen from Figure 3a,b that these areas have a certain amount of energy, which may affect the real targets in the areas.

Because the azimuth ambiguity problem is relative, targets in the ambiguity area also affect the target at (0,0). Figure 4 shows another specific case that there exists a strong scattering target located in the calculated ambiguity area that is received by the antenna sidelobe. Assume the σ of the strong scattering point is 30, and the  σ target at (0,0) is 1. Choosing $\omega t=\frac{\pi }{4}$; the other simulation parameters are the same as in Table 1. From Equation (10), one of strong scattering targets lie in approximately (2161.563,−1271.981,0). Figure 5a shows imaging result at (0,0) just using the signal received from the antenna sidelobe at the ambiguity point. Then, we combine the signal from the antenna main lobe. Figure 5b shows the imaging result. Figure 5c shows the imaging result in the azimuth direction. It can be observed that the sidelobe of the target at (0,0) rises, which causes imaging quality degradation.

### 3.2. Real Data Experiment

These CSAR data were collected by the Aerospace Information Research Institute, Chinese Academy of Sciences, in 2017. Most parameters are very similar to the above simulation parameters, except real PRF is 1179 Hz, trajectory is not an ideal circle, and velocity is not a constant. The experimental scene includes an area whose sea level is roughly zero. Figure 6 shows a schematic part of the trajectory and a general view of part of the experiment scene from Google Earth. 

This article emphasizes the area choice of BP imaging, so we only select a small sub-aperture. For this real data, parameters are designed reasonably, and the fuzzy phenomenon can be avoided by setting limits of beam width. However, we can still study azimuth ambiguity for CSAR using the whole sub-aperture ignoring the beam limit. Figure 7 individually displays the results of imaging on the full reference plane, at the right place, and at one error place. Figure 7b,c uses the same spacing grid. It can be seen that Figure 7c has a fuzzy shape similar to Figure 7b, but the targets are completely defocused. Furthermore, the shape of the collection of fuzzy points in the case of k=1, which is very close to a hyperbolic curve, verifies the equation mentioned in Appendix A to some extent.

In the real data experiment, we can decrease the sampling rate by 10 times to meet extremely poor-quality conditions. From Equation (15) we can get the conclusion that there are at least 40 solutions. From Equations (13) and (14) we can infer that the azimuth ambiguity area is not too defocusing like Figure 7c because the semi-major axis and the semi-minor axis of the ellipse curse are considerably large. Figure 8 shows the normal condition and the under-sampling condition. The experiment’s result shows that the phenomenon is slightly different compared to the azimuth ambiguity phenomenon analyzed by the frequency domain algorithm [13].

## 4. Discussion and Conclusions

This paper introduces a theoretical analysis of ambiguity regions that may occur using the back-projection algorithm. The fundamental analysis is based on the reason that on the reference height plane, there exist points that have a similar distance history to the real targets. This is the difference using the time domain algorithm instead of the frequency domain algorithm. Nevertheless, the traditional method only refers that the system sampling is supposed to satisfy the sampling theorem. In the case of under-sampling, the analysis of the azimuth ambiguity is not mentioned. We discover that due to the geometry of trajectory in CSAR, the azimuth ambiguity area has an exact solution to the ideal point target. The exact solution explains the reason that the ambiguity is defocused and the shape of the ambiguity area on the whole reference height. Furthermore, because the distance history similarity is relative, the solution contributes to the explanation of the effect caused by strong scattering targets received by the antenna sidelobe on the interested targets, which is located in the scene center.

In summary, this article expounds the occurrence of azimuth ambiguity when observing the whole reference height plane using the back-projection algorithm. The work mainly focuses on the analysis of an ideal point target. More complex objects and situations can be discussed following the rules described in the article.

## Figures and Tables

**Figure 1 sensors-19-04920-f001:**
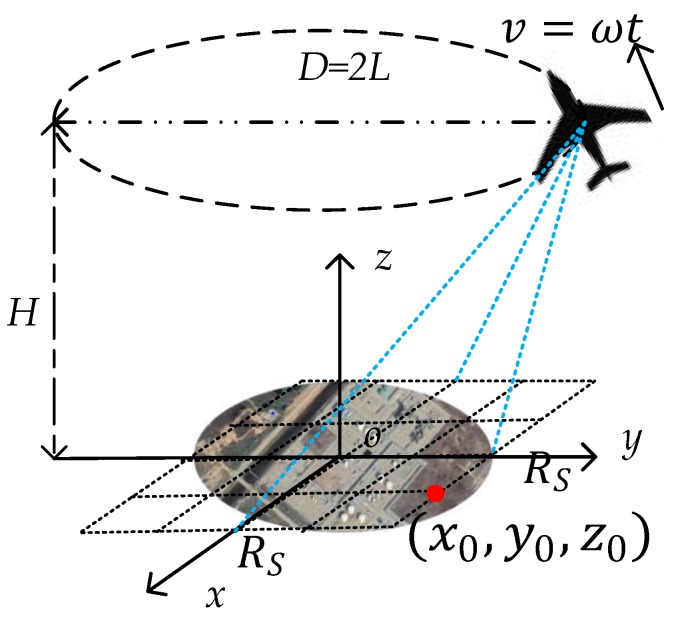
Circular synthetic aperture radar (CSAR) imaging geometry with a Cartesian grid.

**Figure 2 sensors-19-04920-f002:**
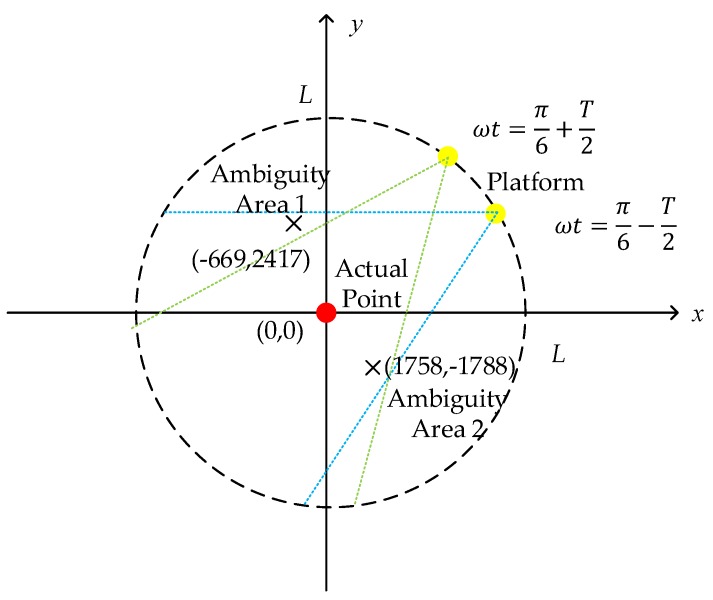
Geometry of the projection on the plane z = 0.

**Figure 3 sensors-19-04920-f003:**
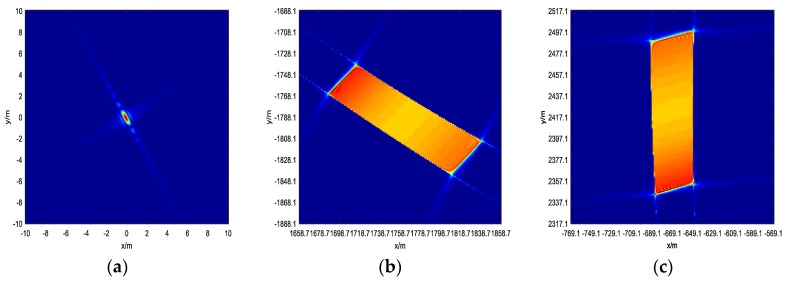
This is the ideal point simulation using the back-projection (BP) algorithm in three areas in the whole reference height plane. (**a**) The center of the area is (0,0,0); (**b**) the center of the area is approximately (1758.696,−1788.093,0); and (**c**) the center of the area is approximately (−669.186,2417.122,0).

**Figure 4 sensors-19-04920-f004:**
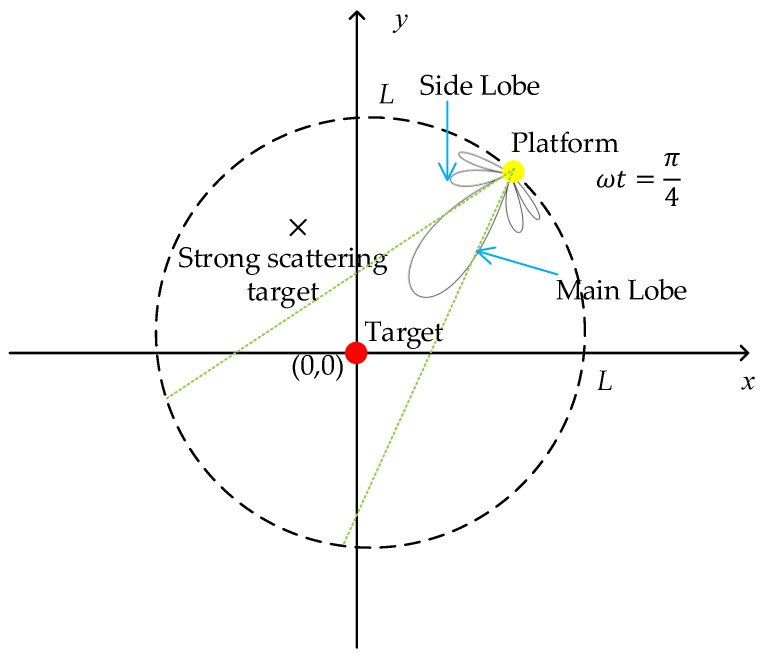
A sketch of a special case. There exists a strong scattering target in the calculated ambiguity area, which falls in the antenna side lobe.

**Figure 5 sensors-19-04920-f005:**
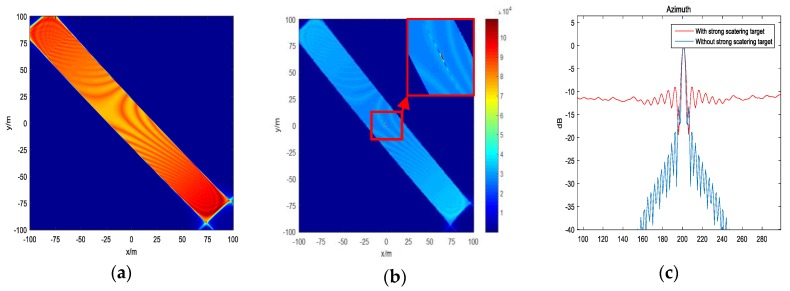
Simulation of this special case: (**a**) Imaging result at point (0,0) only using the signal received from the size lobe; (**b**) imaging result at point (0,0) using the whole signal; and (**c**) imaging result in the azimuth direction.

**Figure 6 sensors-19-04920-f006:**
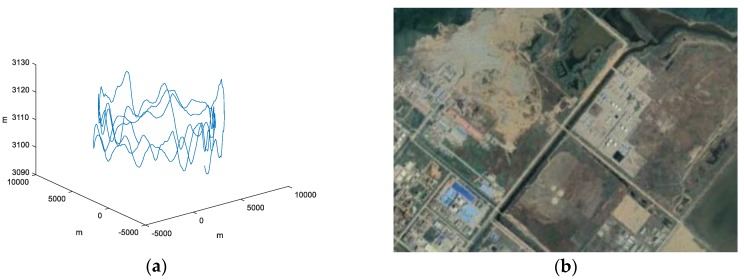
(**a**) The real CSAR trajectory. (**b**) General view of the imaging scene from Google Earth.

**Figure 7 sensors-19-04920-f007:**
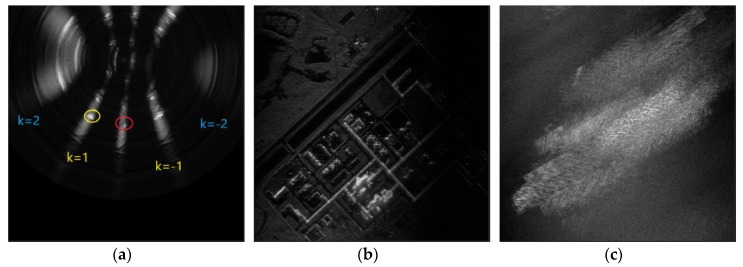
This is the real data experiment. (**a**) BP imaging on the whole reference height plane; (**b**) BP imaging in the right area, which is marked by the red circle in (**a**); and (**c**) BP imaging in one wrong area, which is marked by the yellow circle in (**a**).

**Figure 8 sensors-19-04920-f008:**
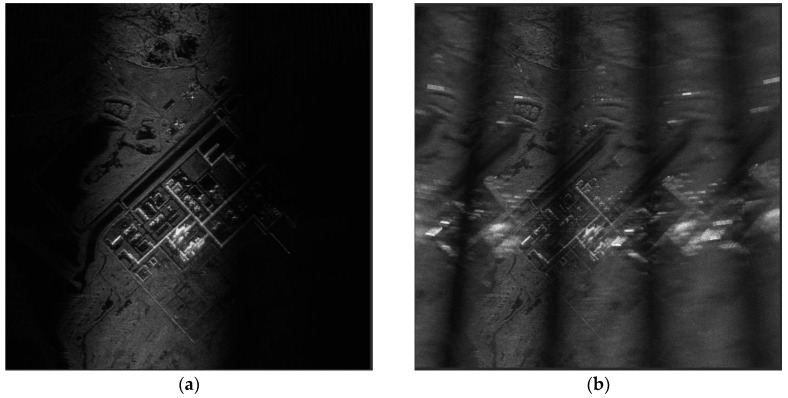
Comparison of normal and under-sampled conditions. (**a**) Original data. (**b**) Processed data after under-sampling.

**Table 1 sensors-19-04920-t001:** Important parameters for simulation.

Parameter	Value
Carrier Frequency	5.4 GHz
Pause Width	20 us
Band Width	560 MHz
Sample Frequency	1.5 GHz
Pause Repeat Frequency(PRF)	1200 Hz
Velocity	80 m/s
Platform Height	3000 m
Radius	5000 m
Sub aperture Size	About 2°
Reference Height	0 m
Slow time	$\omega t=\frac{\pi }{6}-\frac{T}{2}:\frac{\pi }{6}+\frac{T}{2}$

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
