# Peer review of "Analysis of the Azimuth Ambiguity and Imaging Area Restriction for Circular SAR Based on the Back-Projection Algorithm"

_sensors, 2019, doi:10.3390/s19224920_

Round 1
Reviewer 1 Report
This paper mainly focuses on azimuth ambiguity analysis and imaging region limitation analysis of CSAR. Although there are many studies on SAR, there are relatively few in CSAR and WWSAR. Several suggestions for revision are as follows
The line 21, 22, 72... The abbreviation BP, PRF, LFM, should be added to the full name when the first appearance. Above line 74, equation (2) lacks a closing parenthesis. Equation (1) can be derived directly, but equations (2), (3), etc. should either have a derivation process or should have a reference source. Other equations should be the same. In line 118, sinωt, cosωt are not italic, but almost all other places they are italic. The full text should be uniform, it is recommended that sin, cos, tan are not italic, ωt Line 84, the last comma should be a period. The sentence consisting of line 85 and the equation below it is incomplete. Should the "Then" of line 86 be lowercase and not indented? Line 180, “which is marked by the red circle in (b)”. It should be “which is marked by the red circle in (a)”In addition, there are few experimental examples, and experimental data can be supplemented. The state of the art and the comparison with other methods is due in more explicit way.
Reviewer 2 Report
The paper contains theoretical analysis of azimuth ambiguity in CSAR imaging, including analysis of the relationships between azimuth ambiguity and parameters such as PRF, slant range angle, velocity of radar platform. Derivations are verified both on simulated and on real CSAR data.
In the paper I miss explanation of significance of the work presented. As the authors say, azimuth ambiguity is rarely analysed theoretically but the importance of this research should be better justified.
In part 4 (Discussion and conclusion) the authors should focus on the added value of the paper, while they mostly present the reasons for ambiguity (well explained but this explanation should be moved to introduction).
In my opinion after revision the paper can be accepted to Sensors.
Round 2
Reviewer 1 Report
I am satisfied with the revised version.